# Polysaccharide-Based Edible Films/Coatings for the Preservation of Meat and Fish Products: Emphasis on Incorporation of Lipid-Based Nanosystems Loaded with Bioactive Compounds

**DOI:** 10.3390/foods12173268

**Published:** 2023-08-30

**Authors:** Seyed Mohammad Bagher Hashemi, Shima Kaveh, Elahe Abedi, Yuthana Phimolsiripol

**Affiliations:** 1Department of Food Science and Technology, College of Agriculture, Fasa University, Fasa 74616-86131, Iran; hashemi@fasau.ac.ir (S.M.B.H.); e.abedi@fasau.ac.ir (E.A.); 2Faculty of Food Science and Technology, Gorgan University of Agricultural Sciences & Natural Resources, Gorgan 49189-43464, Iran; 3Faculty of Agro-Industry, Chiang Mai University, Chiang Mai 50100, Thailand

**Keywords:** biopolymer, coatings, edible films, meat, nanosystems, encapsulation

## Abstract

The high water and nutritional contents of meat and fish products make them susceptible to spoilage. Thus, one of the most important challenges faced by the meat industry is extending the shelf life of meat and fish products. In recent years, increasing concerns associated with synthetic compounds on health have limited their application in food formulations. Thus, there is a great need for natural bioactive compounds. Direct use of these compounds in the food industry has faced different obstacles due to their hydrophobic nature, high volatility, and sensitivity to processing and environmental conditions. Nanotechnology is a promising method for overcoming these challenges. Thus, this article aims to review the recent knowledge about the effect of biopolymer-based edible films or coatings on the shelf life of meat and fish products. This study begins by discussing the effect of biopolymer (pectin, alginate, and chitosan) based edible films or coatings on the oxidation stability and microbial growth of meat products. This is followed by an overview of the nano-encapsulation systems (nano-emulsions and nanoliposomes) and the effect of edible films or coatings incorporated with nanosystems on the shelf life of meat and fish products.

## 1. Introduction

Meat and fish products are highly nutritious due to their high contents of nutrients such as vitamins, fats, essential amino acids, minerals, and proteins [1,2]. However, their high water and nutritional contents make them susceptible to oxidative deterioration and microbial contamination [2,3]. The most common bacteria involved in meat spoilage are Pseudomonas and Streptococcus, among other classes, and the most common spoilage molds are *Cladosporium* and *Sporotrichum*, among other classes [4,5]. Generally, chemical alteration and microbial growth in meat and fish products lead to quality degradation, wastage, economic loss, and health risks [6,7]. In recent years, numerous innovative food-preservation strategies have been developed to overcome these issues [8].

Food packaging plays a crucial role in food preservation by creating a protective layer around the food product. Food packaging protects the product against any physicochemical, microbial, and environmental damages during transport and storage of food products such as meat and processed meat [9,10,11]. In this regard, edible films or coatings are an effective technology that can improve meat and fish products’ quality and shelf life [12]. They act as good oxygen barriers, prevent moisture transfer (especially lipid-based), and inhibit microbial growth and invasion of pathogenic microorganisms, reducing lipid oxidation and microbial contamination in meat and fish products [13,14,15].

Due to environmental problems, there is currently a global focus on the reduction in waste, in addition to industry demand for cost-effective food preservation methods. As a result, films and coatings from edible biopolymers such as polysaccharides (e.g., alginates, chitosan, pectin, natural gums) and proteins (e.g., zein, whey protein concentrate, collagen, gelatin) have been widely developed [15,16,17,18,19]. Such films or coatings have various advantages over traditional films or coatings from synthetic polymers: edibility, biodegradability, renewability, bioactivity, non-toxicity, being carbon-free or carbon-neutral, reduction in waste, environmentally friendly, and different application methods such as immersion, aspersion, spreading, and brushing (Figure 1) [1,20,21,22]. Additionally, one significant advantage of such films or coatings is the possibility of incorporating various active compounds within their matrix that could even be consumed with the food products, leading to safety enhancement or improvement in nutritional quality and sensory properties [23]. Therefore, the demand for biopolymer-based edible films or coatings incorporated with active compounds has increased.

In recent years, increasing concerns associated with the impact of synthetic compounds on health have led to high demand for natural bioactive compounds [24,25]. However, directly applying bioactive compounds such as essential oils and extracts of plants and fruits in food formulations, films, or edible coatings may adversely affect the sensory quality of food products because of the compounds’ bitter taste and intense flavor [26,27]. Furthermore, their direct use in the food industry faces different obstacles due to their hydrophobic nature, high volatility, and sensitivity to processing, gastrointestinal, and environmental conditions [28,29,30]. They can be degraded by exposure to light, oxygen, and undesirable pH and temperature, causing a reduction or even loss of their bioactive properties [31].

Nanotechnology is a promising method for overcoming these challenges. It is defined as an applied science and technology that aims to develop devices and dosage forms in the range of 1 to100 nm [32,33,34]. Nano-encapsulation technology provides efficient carriers for loading bioactive compounds such as peptides, vitamins, essential oils, and plant extracts, protecting them from undesirable environmental conditions, improving their biological activity, modifying their release profile, and reducing unwanted organoleptic effects [35]. The greater surface area per mass of nanoparticles compared to larger particles with the same chemical composition increases the nanoparticles’ bioactivity, leading to their higher efficiency in increasing the shelf life of foods [25,36]. Moreover, incorporating nanosystems in edible films or coatings improves the gas transport properties, transparency, and mechanical resistance [37,38,39]. It also prevents interactions between the bioactive compound and the food, the biopolymer, or both [25,39,40].

In recent years, different nano-encapsulation systems have been incorporated with edible films or coatings, including nano-emulsions, nanoliposomes, nanofibers, solid lipid nanoparticles, and nanolipid carriers [29,41,42,43]. As previously mentioned, the high sensitivity of meat and meat products to spoilage during storage is a severe challenge in the food industry. However, the high potential of edible films or coatings incorporated with nanosystems in preserving various food products such as vegetables and fresh-cut fruits has been proven in different studies. As a result, the evaluation of their application potential in preserving meat and meat products has received much attention from researchers.

Hence, this work aimed to review the most used nanosystems incorporated with biopolymer-based edible films or coatings for preserving meat and meat products (Figure 2). Their effects on oxidative stability, physicochemical properties, and microbial quality are discussed.

## 2. Polysaccharide Edible Films or Coatings in the Preservation of Meat and Fish Products

Polysaccharide-based edible films or coatings, such as pectin and alginate-based coatings, have received significant attention from researchers for application in food products due to their various advantages, such as high biodegradability, insolubility in non-polar solvents, high solubility in water, non-toxicity, and the ability to form tasteless, odorless, and colorless films or coatings [44,45]. However, they have some disadvantages, including low extensibility and water vapor permeability [46,47]. Nonetheless, different studies reported their combined use with other biopolymers improved their physicochemical properties [48,49].

Various studies have reported using different polysaccharide-based edible films or coatings alone or in combination with other biopolymers in food packaging. Some of these polysaccharides, such as alginate, chitosan, pectin, and starch, are the most available polysaccharides and are commonly used in the food industry. Thus, in the following, we review the application of these polysaccharides alone or in combination with other biopolymers in preserving meat or fish products.

### 2.1. Pectin-Based Films or Coatings

Pectin is a polysaccharide of galacturonic acid units linked by α-(1→4) linkages and is found abundantly in plant cell walls (Figure 3). High concentrations of pectin can be found in the ripened stage of some fruits such as apples and citrus [50,51]. Pectin is produced commercially, having a white to brown color, and is used in the food industry as a gelling, thickening, and stabilizing agent [52,53].

Pectin is a favorable biopolymer for producing edible films or coatings due to its unique properties, including being odorless, non-toxicity, biodegradability, being inexpensive, renewability, and low gas permeability [52,54]. The extensive use of pectin in the food industry faces some challenges due to its high water solubility and hydrophilicity, which results in a poor moisture protective effect [55]. To overcome these challenges, the combination of pectin with other biopolymers is an effective method leading to the improvement in stability and physicochemical properties of pectin-based edible films or coatings [56,57].

Hence, pectin-based edible films or coatings incorporated with bioactive compounds have been used widely for preserving meat and meat products (summarized in Table 1). For instance, Bermúdez-Oria et al. [58] evaluated the effect of edible pectin-fish gelatin films containing the olive antioxidants (hydroxytyrosol and 3,4-dihydroxyphenylglycol) on beef meat during refrigerated storage for 7 days. They stated that the applied edible coating significantly reduced oxidation in terms of TBA value. In this regard, hydroxytyrosol showed more potent antioxidant activity than 3,4-dihydroxyphenylglycol, as at the end of storage time, the TBA value of coated samples containing 0.5% hydroxytyrosol was considerably lower than that of coated samples containing 0.5% 3,4-dihydroxyphenylglycol. The results showed that 0.5% hydroxytyrosol improved lipid stabilization by 68%, while 0.5% 3,4-dihydroxyphenylglycol improved oxidation stability by 59% compared to coated samples without antioxidants. They reported that using edible films containing hydroxytyrosol and beeswax reduced lipid oxidation by 100% during 7 days of storage. They attributed these results to the oxygen barrier property of the used biopolymer-based film and the antioxidant potential of the hydroxytyrosol.

Recently, Guo et al. [59] prepared an active edible film containing pectin and polyphenol of watermelon peel and evaluated its effect on the quality of chilled mutton during super-chilled storage for 35 days. The pectin-based film containing 1.5% polyphenol could significantly control bacterial growth, as the total bacterial counts of active coated samples reached the limit value (7 log CFU/g) after 35 days of storage, while the total bacterial counts of control samples reached the limit value after 21 days of storage. The TBA and TVN values of coated samples were about 0.7 (mg MDA/Kg) and 20 (mg/100 g), respectively, at the end of the storage time (day of 35), which were significantly lower than control groups. They stated that the oxidative stability of coated samples could be related to the low storage temperature, the inhibition effect of watermelon extracts on the growth of the microorganisms, and the decomposition reduction of nitrogen-containing macromolecules and proteins. Generally, they attributed the preservative effect of active film to the polyphenol compounds of watermelon peel, such as chlorogenic acid, myricetin, and caprylic acid.

### 2.2. Alginate Based Films or Coatings

Alginate is a non-branched polysaccharide from brown seaweeds, composed of β-D-mannuronic acid (M) and α-L-guluronic acid residues (G) (Figure 4) [18,66]. The sequence of G and M residues of alginate depends on its sources. It has been reported that the G content of alginate extracted from algae, L. *hyperborean* stems is about 60%, while the G content of alginates extracted from other commercial algae species is about 14–31% [67]. Generally, alginate forms a hydrogel by crosslinking with divalent or multivalent ions, and the carboxylate groups of its G residues crosslink with such ions to form a 3D network [68].

Alginate is a biocompatible and non-toxic polysaccharide, and is known as an effective biopolymer for the production of edible films or coatings due to its distinctive functions as a stabilizing, film and gel-forming, and thickening agent [69,70]. It has different advantages such as renewability, biodegradability, biocompatibility, non-toxicity, and low cost [71,72]. It has been reported that the chemical composition of alginate (the molecular weight, the proportion of G residues to M residues, and the length of G-blocks) mainly affects its hydrogels’ physical properties [73,74]. In recent years, alginate-based edible films or coatings have received significant attention for extending the shelf life of food products such as meat and meat products due to alginate’s high potential in controlling permeability, reducing dehydration, and improving the mechanical properties of edible films or coatings [75]. Despite the mentioned advantages, low stability under wet conditions and the high sensitivity to degradation processes were identified as the disadvantages of alginate-based edible films or coatings [76]. It has been reported that combining alginate with other polymers, such as protein [77], chitosan [78], or cellulose [79], is an effective modification to overcome these disadvantages.

Recent studies of the effect of alginate-based edible films or coatings are reviewed in Table 2. Heydari et al. [80] investigated the effect of alginate-based edible coating containing horsemint essential oil on the quality of bighead carp fillets during storage at 4 °C. The results of this study showed that the applied edible coating enriched with the horsemint essential oil could significantly reduce the spoilage of fillets and enhance their shelf life by increasing oxidation stability and reducing microbial deterioration. The wrapped sample with an alginate-based edible coating enriched with 1% horsemint essential oil had the lowest TVN, TBA, peroxide, and free fatty acid at the end of the storage time. The total viable counts and total psychrotrophic counts of the uncoated sample exceeded the maximum acceptable limit of 6 log CFU/g after 8 days of storage, but the coated sample containing 1% horsemint essential oil exceeded the limit after 12 days of storage. They attributed this considerable microbial reduction to the role of the coating as an effective barrier to oxygen transfer, leading to growth inhibition of the aerobic bacteria. In another study, Golpaigani et al. [81] investigated the effect of alginate–chia-seed-based edible coating containing rainbow trout roe protein hydrolysate on the quality of fresh meat during storage at 4 °C. They revealed that the applied coating could considerably control lipid oxidation and microbial growth, and, by increasing the concentration of rainbow trout roe protein hydrolysate, better results were achieved. It should be noted that in most cases the mentioned coating had better function than the coating containing BHA.

### 2.3. Chitosan-Based Films or Coatings

Chitosan is a polymer (Figure 5) obtained from crustacean shells by deacetylation of chitin under alkaline solutions [85]. Chitosan is a non-toxic, safe, renewable, biodegradable, allergen-free, eco-friendly, and biocompatible polymer with health-beneficial properties. It is soluble in acidic pH but highly insoluble in water [86,87]. In addition, chitosan has effective antibacterial activity because of its positive charge. It has been suggested that the primary mechanism of the antibacterial property of chitosan is the electrostatic interactions between the anionic groups of the bacterial cell membrane and the cationic structure of chitosan, leading to membrane permeability and leakage of biological components of the bacterial cell and, finally, cell death [88,89]. However, chitosan has some disadvantages, such as low solubility in neutral and alkaline pH and low mechanical resistance [90].

Furthermore, chitosan exhibits considerable film-forming properties, which attracted researchers’ attention for its application in edible packaging [91]. In recent years, the suitable structure properties of chitosan have made it an attractive choice for producing edible films or coatings. Chitosan films or coatings are effective barriers against oxygen and carbon dioxide transfer, which can preserve the quality of the wrapped foods and extend their shelf life [92]. It has been proven that the incorporation of chitosan with bioactive compounds such as natural essential oils and extracts makes them effective active films or coatings for the preservation of meat and fish products (as summarized in Table 3). Fang et al. [93] investigated the effect of chitosan coating incorporated with gallic acid on the quality of fresh pork in modified atmosphere packaging. They stated that the incorporation of gallic acid in chitosan edible coating could significantly inhibit microbial growth during 20 days of storage at 4 °C, but no significant differences were observed between coated samples containing 0.2 and 0.4 gallic acid (*p* > 0.05). In addition, the mentioned samples had the lowest TBA value compared to the control (uncoated) and chitosan-coated sample without gallic acid. Conversely, the gallic acid concentration had a significant effect on protein oxidation, as the coated sample containing 0.4% gallic acid showed the lowest free thiol group values. Xiong et al. [94] evaluated the effect of a chitosan–gelatin-based edible coating containing nisin and grape seed extract on the quality of fresh pork during cold storage at 4 °C for 20 days. They reported that the coated samples incorporating 0.5 grape seed extract and 0.5 grape seed extract + 0.1 nisin showed the lowest TBA value at the end of the storage time. They attributed these findings to the high antioxidant potential of phenolic compounds in grape seed extract. Regarding the total viable counts, the coating could significantly control the microbial growth of samples, even without grape seed extract or nisin.

### 2.4. Starch-Based Films or Coatings

Starch is a biopolymer in the plant storage organs (endosperm), such as roots, legumes, unripe fruits, and tubers [100]. It is considered to be one of the most important natural polysaccharides to replace plastics due to its unique properties such as low cost, thermoplastic nature, low permeability to oxygen, easy access, and high resolution [101,102]. There are two types of starch molecules: amylose (a linear polymer in which D-glucose units are linked by α-1,4 glucosidic bonds) and amylopectin (a branched polymer in which the main chain is linked to the branches via α-1,6 glucosidic bonds) [103]. The type of starch and the ratio of amylose/amylopectin affects the properties of films or coatings, such as color, thickness, moisture content, and mechanical properties [104]. Amylose has a helical structure and is responsible for the development of films, and starch-based films having a higher amount of amylose have better film characteristics, such as plasticity, gas barrier properties, and mechanical strength [105,106]. Generally, the properties of starch-based edible films and coatings depend on the type of applied starch. Various studies evaluated the effect of starch edible films or coatings on the preservation of meat and fish products (Table 4). In this regard, research was conducted by Yıldırım-Yalçın et al. [107] on the effect of crosslinked maize starch edible film incorporated with grape juice on the quality of chicken breast fillets during storage. They revealed that the coating significantly reduced the moisture loss of samples during all days of storage. This was the result of covering the surface of chicken samples, which reduced the moisture transfer between the atmosphere and the sample. In addition, the coating reduced the growth of mesophilic aerobic, psychrophilic, and Enterobacteriaceae bacteria by 1.58, 0.94, and 0.85 log, respectively, after 8 days of storage.

Asyila Marzlan et al. [108] evaluated the effect of starch-based edible films incorporating torch ginger inflorescence essential oil on the preservation of chicken meat. The addition of torch ginger inflorescence essential oil significantly affected the mechanical properties of starch films. The tensile strength decreased by adding 0.1 and 0.2% essential oils, while higher concentrations (0.4 and 0.8%) increased the edible films’ tensile strength. In addition, the results showed that the antioxidant potential of starch-based edible films increased by increasing the concentration of essential oil in terms of DPPH and ABTS radical scavenging activity. Moreover, they revealed that the active edible film significantly reduced the samples’ weight loss and limited the samples’ pH increase during 6 days of storage. At the end of the storage time, the TBA value of all packed samples was lower than that of uncoated ones, and the lowest value (0.2 mg MDA/Kg) was from coated samples with starch-based films incorporated with essential oil. The antioxidant compounds of torch ginger inflorescence essential oil react with free radicals and convert them to more stable molecules, inhibiting lipid oxidation. In addition, starch-based active film significantly limited the growth of coliforms during 6 days of chilled storage, and coated samples with starch-based films incorporated with essential oil had the lowest coliform counts (4.98 ± 0.07 log CFU/g) at the end of the storage time in comparison to uncoated samples.

## 3. Biopolymer Edible Films or Coatings Incorporated with Nanosystems in the Preservation of Meat and Fish Products

The direct use of bioactive compounds in edible films or coatings faces some challenges due to their sensitivity to oxygen, light, and heat. In addition, the interaction between the bioactive compounds and the components of edible films or coatings can change the bioactive compounds’ activity and modify the chemical structure of films or coatings and, subsequently, their functional properties [117,118]. In this regard, nanotechnology is an innovative technology that is a promising means of overcoming these challenges. Various studies showed that different nano-encapsulation systems, such as nanoliposome and nano-emulsion, are effective for applying bioactive compounds in edible films or coatings. In the following, we review the application of edible films or coatings incorporated with nano-emulsion and nanoliposomes in preserving meat and fish products.

### 3.1. Biopolymer Edible Films or Coatings Incorporated with Nano-Emulsions

Emulsions are categorized into microemulsions (4–200 nm), nano-emulsions (20–200 nm), and macroemulsions (200 nm–200 µm) based on their particle size, which affects their function [119]. Generally, nano-emulsions are defined as colloidal systems consisting of oil and aqueous phases, and there are two kinds of nano-emulsions according to the phases: oil/water (o/w) or water/oil (w/o) [25]. Nowadays, nano-emulsion-based edible films or coatings are used increasingly in food preservation. The o/w nano-emulsions are preferred for application in edible films or coatings due to their suitable capability to incorporate various lipophilic bioactive compounds into the hydrophilic biopolymer matrix [25]. The efficiency of nano-emulsions in edible films or coatings depends on the particle size of emulsions, their release behavior, and their stability during storage time. The desirable particle size can be achieved by homogenization, which leads to a reduction in the particle size of emulsions through mechanical sheer force. Nano-emulsions are kinetically stable but thermodynamically unstable; surfactants are used to reduce the interfacial tension between the dispersed phase and the dispersed medium of nano-emulsions [120,121]. Generally, nano-emulsions are effective delivery systems for bioactive compounds such as natural extracts and essential oils, and have great potential for application in edible films or coatings [122,123]. In the following, we review some recent studies about the effect of edible films or coatings incorporating nano-emulsions in the preservation of meat and fish products (summarized in Table 5). Xiong et al. [27] investigated the effect of a pectin edible coating incorporating oregano essential oil and resveratrol nano-emulsion on the preservation of pork loin in modified atmosphere packaging. They revealed that the edible coating could significantly extend the shelf life of samples by retarding lipid and protein oxidation, maintaining meat tenderness, and inhibiting microbial growth. They stated that the coated sample containing microemulsion had a lower TBA value compared to the coated sample containing nano-emulsion at the end of the storage time. They attributed this finding to the weak ability of the coating with a smaller particle size to inhibit the lipid oxidation.

### 3.2. Biopolymer Edible Films or Coatings Incorporated with Nanoliposome

Nanolipid-based carriers such as nanoliposomes are known to be effective carriers to enhance stability and solubility and control the release of bioactive compounds such as natural extracts and bioactive peptides [132]. It has been reported that nanoliposome systems are biodegradable, non-toxic, and have the suitable ability for loading both hydrophilic and lipophilic compounds [133]. Despite all the advantages of nanoliposomes, it has been reported that their direct application in food formulations can lead to rupture or aggregation of the system; also, the leakage of encapsulated bioactive compounds may occur [134]. Thus, edible films or coatings are considered suitable carriers for loaded nanoliposomes to be applied in food preservation. In this regard, some studies evaluated the effect of edible films or coatings incorporated with nanoliposomes on the shelf life of meat and fish products. Cui et al. [135] investigated the effect of a chitosan coating incorporated with a phage-loaded nanoliposome on *E. coli* in beef. They reported that the encapsulation efficiency and particle size of the loaded nanoliposome was 57.66% and 150.1 nm, respectively. The sample coated with chitosan film containing the liposome-encapsulated phage had the lowest *E. coli* after 15 days of storage. Cui et al. [42] evaluated the effect of xanthan gum-based edible coating incorporated with *Litsea cubeba* essential oil nanoliposomes in the preservation of salmon. They stated that the particle sizes of nanoliposomes were in the range of 149.92–185.39 nm. They reported that increasing the concentration of *Litsea cubeba* essential oil from 4 to 7 mg/mL increased the particle size but significantly reduced the encapsulation efficiency. The TBA values of coated samples were significantly lower than that of the control and the lowest value was related to the coated sample containing liposome: xanthan gum in the proportion of 1:3.

## 4. The Safety of Edible Films or Coatings Incorporated with Nano-Encapsulated Bioactive Compounds

Food safety is crucial in food legislation [136]. In this regard, edible films or coatings incorporated with nano-encapsulated bioactive compounds in packaging technology should be safe in terms of health and without any adverse effects on consumers’ health. In other words, since they are an edible part of wrapped food products, they directly impact consumers; hence, they should follow the required regulations for food ingredients [137]. In this regard, all used components of edible films or coatings, as well as their bioactive additives such as natural essential oils, extracts, and bioactive peptides, should be GRAS and be approved by the FDA [138]. In addition, all ingredients applied in the edible films or coating should be clearly declared on the label, especially as some can be allergens for some consumers [139].

The toxicity effect of nanomaterials is another significant aspect of the safety of edible films or coatings incorporated with nano-encapsulated compounds. Experimental studies showed that nanomaterials have the potential to cause inflammatory and toxic effects depending on their chemical composition, size distribution, and shape. In addition, the usage doses have a considerable impact on their toxicity potential [140,141].

Scientists believe that nanomaterials with smaller sizes have more toxicity potential. It has been reported that nanomaterials’ chemical composition and size have the most significant effect on their toxicity toward humans. In this regard, studies showed that nanoparticles with a size less than 100 nm penetrate cells, and those smaller than 35 nm can cross the blood–brain barrier. Moreover, smaller nanoparticles have higher catalytic activity, which may produce more reactive oxygen species, higher adsorption rates, and binding ability, and finally affect residence times inside the human body [141,142,143].

FDA guidelines for using nanomaterials in food products suggest that a change in the production of the materials via nanotechnology may have considerable effects on their essence, safety, and supervision status. Thus, the FDA suggests that safety evaluations of the products can be consistent with information obtained by the nanometer version [144]. In other words, nanoparticles exhibit different biological and physicochemical properties compared to larger materials, so the evaluation of their inherent safety, either alone or in food products, is a critical issue from scientists’ perspectives. This led to the provision of various opinions by the FDA and EFSA on the adequacy of the risk evaluation guidelines for the industry concerning the safety evaluation of nanoparticles [144,145,146].

In this regard, the FDA provided a guidance document entitled “The evaluation of whether the product adjusted according to FDA entails nanotechnology or not” and intends to enforce the stated policies. It should be noted that the FDA has declared some nanomaterials, such as nano-clay, aluminum, carbon, and zinc oxide. Generally, information certifying that the product is made in compliance with the FDA guidance document should be provided by the manufacturer for the users [147].

## 5. Conclusions

Due to increasing demand for food products having minimum processing, researchers’ attention has been attracted to active packaging. In recent years, due to environmental problems, global demand for waste reduction, and industry demand for cost-effective food preservation methods, edible films or coatings have gained significant attention and are used as alternatives to synthetic polymers due to their eco-friendly and non-toxic nature. In this regard, different studies have evaluated the effect of active edible films or coatings from biopolymers, such as polysaccharides and proteins, on the shelf life of food products, especially highly perishable ones such as meat and fish products. However, there are some challenges in the direct use of bioactive compounds in edible films or coatings; they can have undesirable effects on organoleptic food products’ properties, and their sensitivity to adverse conditions makes them unstable and reduces their bioactivity. Nanotechnology is a promising method for overcoming these challenges. Nano-encapsulation technology provides efficient carriers for loading bioactive compounds such as peptides, essential oils, and plant extracts, protecting them from undesirable environmental conditions and improving their biological activity. The greater surface area per mass of nanoparticles compared to larger particles with the same chemical composition increases the bioactivity of nanoparticles. Moreover, incorporating nanosystems in edible films or coatings prevents interactions between the bioactive compound and the food, the biopolymer, or both. It has been reported that different nanosystems, such as nano-emulsions and nanoliposomes, have the potential to be used in edible films or coatings to extend the shelf life of food products by inhibiting microbial growth and increasing protein and lipid oxidation stability. As reviewed in this paper, biopolymer-based edible films or coatings incorporated with nanosystems are effective choices for the preservation of meat and fish products without the application of synthetic preservative compounds. Finally, despite the studies reviewed here, identifying cost-effective, safe, and effective edible films or coatings with high consumer acceptability will require significantly more work. In particular, the behavior of active films or coatings incorporated with nanosystems after consumption should be clear, to be safe and without any undesirable effects.

## Figures and Tables

**Figure 1 foods-12-03268-f001:**
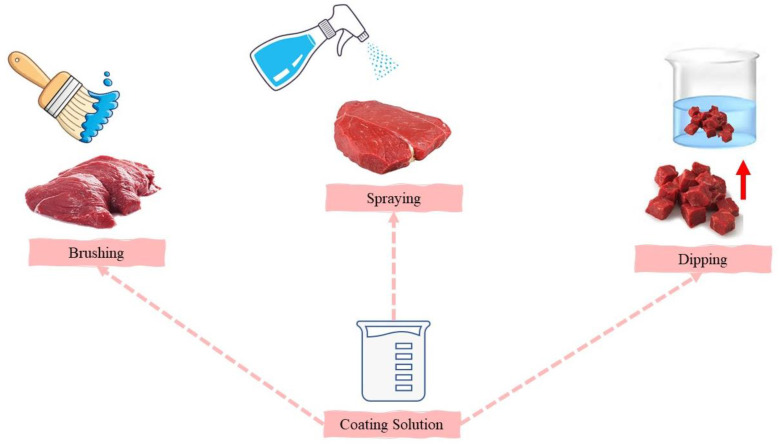
Some methods of applying edible coating.

**Figure 2 foods-12-03268-f002:**
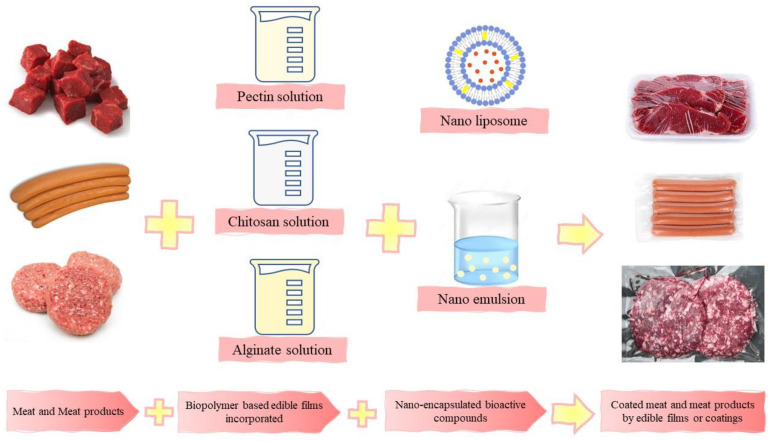
Preservation of meat and meat products by biopolymer-based edible films or coatings incorporated with nano-encapsulated bioactive compounds.

**Figure 3 foods-12-03268-f003:**
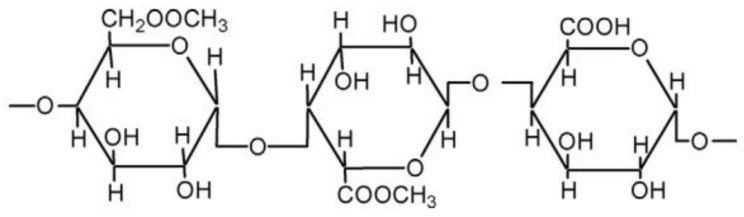
Chemical structure of pectin.

**Figure 4 foods-12-03268-f004:**
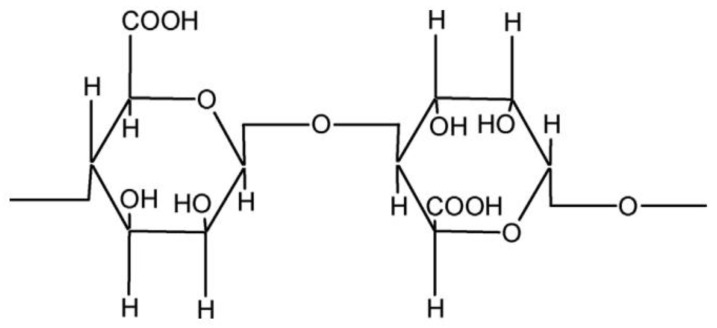
Chemical structure of alginate.

**Figure 5 foods-12-03268-f005:**
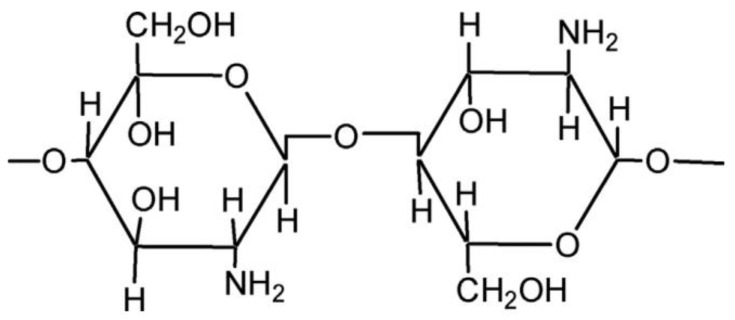
Chemical structure of chitosan.

**Table 1 foods-12-03268-t001:** The effect of pectin-based edible films or coatings on the preservation of meat and fish products.

Type of Biopolymer	Applied Bioactive Compound	Method of Application	Coated Product	Results	References
Citrus pectin (with an esterification degree of 53%) + fish skin gelatin	Hydroxytyrosol and 3,4 dihydroxyphenylglycol from olive oil by product	-	Beef meat	➢ Improve lipid stabilization by 68% and 59% of coated samples containing 0.5% hydroxytyrosol and 3,4 dihydroxyphenylglycol, respectively	[58]
Pectin of watermelon peel	Polyphenols of watermelon peel	Covering	Chilled mutton	➢ Reduction in TVN and TBA value of coated samples containing 1.5% polyphenols of watermelon peel ➢ Reduction in total bacterial counts	[59]
Citrus pectin	Green tea powder	Dipping	Pork patty	➢ Reduction in TBA value during 14 days of storage ➢ Inhibition in total plate counts	[60]
Pectin	Sodium lactate and sodium diacetate	Applying 300 µL of coating solution on each side of the salmon discs	Cold smoked salmon	➢ Reduction in growth of *L. monocytogenes*	[61]
High-methoxyl pectin	*Thymus vulgaris* and *Thymbra spicata* essential oils or extracts	Dipping	Sliced bolognas	➢ A 1.73 log CFU/g reduction in microbial counts of coated samples containing 0.5% *Thymus vulgaris* and 0.5% *Thymbra spicata* essential oils ➢ Reduction in TBA value of coated samples containing 0.5% essential oils and extracts	[62]
Pectin	Oregano essential oil	Immersing	Large yellow croakerfillet	➢ Reduction in the growth of microorganisms (total viable, *Pseudomonas* bacteria and H2S-producing bacteria counts) ➢ Reduction in TBA and TVN values	[63]
Sodium alginate + pectin	Cinnamic acid	Wrapping	Fresh beef loins	➢ Inhibition in bacterial growth	[64]
Pectin (with an esterification degree of 70%) + fish skin gelatin	Lemongrass essential oil	Wrapping	Chicken meat	➢ A slight change in pH value during 6 days of storage	[65]

**Table 2 foods-12-03268-t002:** The effect of alginate-based edible films or coatings on the preservation of meat and fish products.

Type of Biopolymer	Applied Bioactive Compound	Coated Product	Results	References
Sodium alginate	Sodium lactate and sodium diacetate	Cold-smoked salmon	✓ Reduction in growth of *L. monocytogenes*	[61]
Sodium alginate	Horsemint essential oil	Bighead carp (*Aristichthys nobilis*) fillets	✓ Reduction in lipid oxidation of the coated sample containing 1% horsemint essential oil in terms of TBA, TVN, peroxide, and free fatty acid values ✓ Reduction in the total viable counts and total psychrotrophic counts by about 4.5 and 2.5 logs respectively in comparison to the uncoated sample	[80]
Sodium alginate + chia gum	Protein hydrolysate of rainbow trout roe	Fresh meat	✓ Reduction in total viable counts of the coated sample containing 1.5% protein hydrolysate by about 3 logs compared to the non-coated sample ✓ Reduction in lipid oxidation in terms of TBA, peroxide, and TVN values	[81]
Sodium alginate	Propionic acid and thyme essential oils	Fresh chicken breast fillets	✓ Reduction in total viable counts of the coated sample containing propionic acid by about 4 logs compared to the control sample ✓ Reduction in lipid oxidation in terms of TBA value until the third day of storage	[82]
Sodium alginate + maltodextrin	*Asparagus racemosus* extract	Chevon sausages	✓ Reduction in lipidoxidation of the coated sample containing 2% *Asparagus racemosus* extract in terms of TBA and free fatty acid value ✓ Reduction in total plate counts, psychrophilic count, and yeast and mold count	[83]
Sodium alginate	Bacteriophage ϕIBB-PF7A	Chicken breast	✓ Inhibition of the growth in *P. fluorescens*	[84]

**Table 3 foods-12-03268-t003:** The effect of chitosan-based edible films or coatings on the preservation of meat and fish products.

Type of Biopolymer	Applied Bioactive Compound	Coated Product	Results	References
Chitosan with 75%–90% deacetylation + fish collagen from blue shark skin	-	Red porgy meat	✓ Reduction in lipid oxidation in the coated sample in terms of TVN and K values	[20]
Chitosan with >90% deacetylation	Gallic acid	Fresh lion pork	✓ Reduction in total viable counts in the coated sample containing 0.2 and 0.4 chitosan ✓ Reduction in TBA value ✓ Reduction in protein oxidation	[93]
Chitosan with >90% deacetylation +gelatin	Nisin + grape seed extract	Fresh pork	✓ Reduction in lipid oxidation in the coated sample containing grape seed extract in terms of TBA value	[94]
Chitosan with 85% deacetylation	-	Silver carp	✓ Reduction in total viable counts ✓ Reduction in lipid oxidation in terms of TVN and TBA values	[95]
Chitosan	Grape seed extract + tea polyphenols	Red drum fillets	✓ Reduction in lipid oxidation in terms of TVN and TBA value ✓ Reduction in total viable counts	[96]
Chitosan with medium molecular weight	*Zataria multiflora* Boiss oil + sumac extract	Beef steaks	✓ Reduction in total viable cell counts, lactic acid bacteria, *Enterobacteriaceae*, *Pseudomonas,* and yeasts–molds in coated samples containing 4% sumac extract and 1% *Zataria multiflora* Boiss oil ✓ Reduction in oxidation in terms of TBA and peroxide values	[97]
Chitosan with > 90% deacetylation +salmon fish bone gelatin	Gallic acid + clove oil	Fresh salmon fillet	✓ Reduction in lipid oxidation in the coated sample without bioactive compounds and coated sample containing gallic acid and clove oil ✓ Reduction in total viable cell counts in the coated sample containing gallic acid and clove oil	[98]
Chitosan with molecular weight of 340 and 75–85% deacetylation	Grape seed extract + *Origanum vulgare* essential oil	Turkey breast meat	✓ Reduction in lipid oxidation in the coated sample containing 2% grape seed extract and 1% *vulgare* essential oil in terms of TBA and TVN values ✓ Reduction in total viable cell counts, lactic acid bacteria, *Enterobacteriaceae*, *Pseudomonas,* and yeasts–molds	[99]

**Table 4 foods-12-03268-t004:** The effect of starch-based edible films or coatings on the preservation of meat and fish products.

Type of Biopolymer	Coated Product	Applied Bioactive Compound	Results	References
Maize starch	Chicken breast fillets	Grape juice	✓ Reduction in total aerobic mesophilic, psychrophilic, and Enterobacteriaceae counts by 1.58, 0.94, and 0.85 log, respectively ✓ Reduction in TBA value by 1.07 mg MDA kg^−1^	[107]
Starch	Chicken meat	Torch ginger inflorescence essential oil	✓ Reduction in coliform growth ✓ Reduction in lipid oxidation in terms of TBA value	[108]
Chitosan–tapioca starch	Salmon slice	Potassium sorbate	✓ Chitosan solution was the best coating in reduction in aerobic mesophilic and psychrophilic cell counts ✓ Chitosan–tapioca starch-based films led to the reduction in *Zygosaccharomyces bailii* external spoilage but were not effective against *Lactobacillus* spp.	[109]
Corn starch	Raw beef	Clove and cinnamon essential oils	✓ Reduction in microbial populations	[110]
Cassava-starch	Ground beef meat	Lemongrass essential oil	✓ Reduction in total microbial counts of the coated sample with an edible film containing 4% lemongrass essential oil	[111]
Cassava starch	Cooked ham	Chitosan and gallic acid	✓ Reduction in total microbial counts during 28 days of storage in coated sample containing 25 and 150 mg chitosan/g starch	[112]
corn starch-chitosan	Fresh beef Slices	Pomegranate peel extract and *Thymus kotschyanus* essential oil	✓ Reduction in total microbial counts and lactic acid bacteria counts during 21 days of storage ✓ Reduction in lipid oxidation in term of TBA value during 21 days of storage	[113]
Cassava Starch/Gelatin	Sliced pork meat	Quercetin and TBHQ	✓ Decrease in redness reduction during 12 days of storage	[114]
Starch	Beef steaks	Oregano, clove leaf, and rosemary oils	✓ Reduction in TBA value and the best results obtained in coated sample containing 1% oregano oil and 1% clove oil	[115]
Ginger starch	Ground beef	Coconut shell liquid smoke	✓ Reduction in *E. coli* populations of coated samples with an edible film containing 15% coconut shell liquid smoke by 1.33 log cfu/g during 12 days of storage at 4 °C ✓ Reduction in lipid oxidation in terms of TBA value	[116]

**Table 5 foods-12-03268-t005:** The effect of edible films or coatings incorporated with nano-emulsions on the preservation of meat and fish products.

Type of Biopolymer	Applied Bioactive Compound	Nano-emulsion Properties	Coated Product	Results	References
Pectin	Oregano essential oil and resveratrol	Particle size of 53.09 nm PDI of 0.21	Pork loin	✓ Reduction in lipid oxidation in terms of TBA value ✓ Reduction in protein oxidation	[27]
Sodium caseinate	Ginger essential oil	Particle size of 57.4 nm PDI of 0.22 Zeta potential of −18.7 mV	Chicken breast fillet	✓ Reduction in total aerobic psychrophilic bacteria in coated sample containing 6% ginger essential oil during 12 days of storage	[124]
Chitosan	* Zataria Multiflora * Boiss and *Bunium persicum* Boiss essential oils	Particle size of 130.2 and 154.26 nm PDI of 0.28 and 0.24	Turkey meat	✓ Reduction in growth of total viable bacteria, total psychrophilic, *Pseudomonas*, *Enterobacteriaceae*, lactic acid bacteria, and yeast and mold counts in coated sample containing 1% nano-emulsion of *Zataria multiflora* Boiss	[125]
Basil seed gum	Shirazi thyme and summer savory essential oils	-	Chicken fillets	✓ Reduction in lipid oxidation in terms of TVN, TBA and peroxide values ✓ Reduction in total viable count, psychrotrophic and lactic acid bacteria counts	[126]
Glucomannan/carrageenan	Camellia oil	-	Chicken meat	✓ Reduction in total viable counts of coated samples containing 2.5–3.5% camellia oil after 10 days of storage ✓ Reduction in lactic acid bacteria of coated samples containing 3 and 3.5% camellia oil after 10 days of storage ✓ Reduction in lipid oxidation in terms of TBA and TVN values	[127]
Alginate	* Trachyspermum ammi * essential oil	Particle size of 73.5 nm PDI of 0.441	Turkey fillets	✓ Prevention of the growth in *L. monocytogenes* even after 12 days of storage	[128]
Gelatin and chitosan	Rosemary extract and ε-poly-L-lysine	Particle size of 257 nm and 1122.44 nm for the nano-emulsion and a coarse emulsion, respectively Zeta potential of 32.50 and 26.25 mV for the nano-emulsion and a coarse emulsion, respectively	Ready-to-eatcarbonado chicken	✓ Reduction in growth of total microbial counts and yeast and molds ✓ Reduction in TBA value during 16 days of storage	[129]
Chicken bone gelatin-chitosan	Cinnamon essential oil and rosemary extract	Particle size of 183.60 nm and 1370.83 nm for the nano-emulsion and a coarse emulsion, respectively Zeta potential of 33.76 and 25.76 mV for the nano-emulsion and a coarse emulsion, respectively	Ready-to-eat chicken patties	✓ Reduction in total viable counts ✓ Reduction in TBA and TVN values	[130]
Chitosan	Nisin and carvacrol	Particle size in the range of 1741–3893 nm, PDI in the range of 0.095–0.675, and zeta potential in the range of 47–54.77 at different concentrations of NaCl and polyglyceryl-6-dioleate	Salmon fillets	✓ Reduction in total viable counts ✓ Reduction in TBA and TVN values during 15 days of storage	[131]

## Data Availability

The data used to support the findings of this study can be made available by the corresponding author upon request.

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
