# Peer review of "Polysaccharide-Based Edible Films/Coatings for the Preservation of Meat and Fish Products: Emphasis on Incorporation of Lipid-Based Nanosystems Loaded with Bioactive Compounds"

_foods, 2023, doi:10.3390/foods12173268_

Round 1

Reviewer 1 Report

1.      The paragraphs in the introduction are too long. Please divided them into small paragraphs, each portraying information about one key topic.

2.      Authors are suggested to include a figure in the introduction that depicts a comprehensive overview of the topic.

3.      Please provide the regulatory aspects of these edible coating systems.

4.      Please discuss the challenges as well as advantages associated with each kind of coating system.

5.      Please includes some figures that gives through information about the different coating mechanisms.

6.      Please divide the conclusion section into two paragraphs one providing the summary of the existing information and the other providing the future outlook.

Please correct the grammatical errors.

Author Response

Reviewer 1

  1. The paragraphs in the introduction are too long. Please divided them into small paragraphs, each portraying information about one key topic.

Ans: The paragraphs in the introduction have been divided into small paragraphs.

  1. Authors are suggested to include a figure in the introduction that depicts a comprehensive overview of the topic.

Ans: Figure 1 and 2 have been added.

  1. Please provide the regulatory aspects of these edible coating systems.

Ans: The safety and regulatory aspects have been added as presented in Section 4 (The safety of edible films or coatings incorporated with nano-encapsulated bioactive compounds).

  1. Please discuss the challenges as well as advantages associated with each kind of coating system.

Ans: The advantages and disadvantages of each coatings systems have been mentioned.

  1. Please includes some figures that gives through information about the different coating mechanisms.

Ans: The figure about the coating mechanisms has been added.

  1. Please divide the conclusion section into two paragraphs one providing the summary of the existing information and the other providing the future outlook.

Ans: The conclusion is divided into separate paragraphs.

Comments on the Quality of English Language, please correct the grammatical errors.

Ans: All grammatical errors have been edited.

Reviewer 2 Report

The authors tried to review edible coatings in the preservation of meat in the nano-encapsulated natural bioactive aspects. The emphasis is interesting, but the review is not comprehensive.

1. The references should be updated with more recent literature, especially in the last year. In line 216, the author stated "Recently, Xiong et al. ...". We can't say a study in 2020 as recently. As a review, we can't just put the literatures in our endnote together, we have to update and put forward some points to readers. 

2. Based on the title of the review, I would expect news about nanoencapsulation and bioactives. But most of the review is about biopolymer (cabohydrates) coating on meat. The authors should more focus on bioactives. There are a lot reviews about biodegradable package on food or meat, what's new the review could provide to readers? The authors should think about it.

3. I suggest the authors should add a part about the type of the bioactives encapsulated within edible coating. Which kind of bioactive are commonly used and how do they encapsulated by various methods. 

4. I suggest the authors add a schematic illustration to show the method of bioactive nano-encapsulated edible film making. I think there are different methods for the film making in literature. As a reader, I think this part will help the other researchers. 

Author Response

Reviewer 2

The authors tried to review edible coatings in the preservation of meat in the nano-encapsulated natural bioactive aspects. The emphasis is interesting, but the review is not comprehensive.

  1. The references should be updated with more recent literature, especially in the last year. In line 216, the author stated "Recently, Xiong et al. ...". We can't say a study in 2020 as recently. As a review, we can't just put the literatures in our endnote together, we have to update and put forward some points to readers.

Ans: It has been corrected.

  1. Based on the title of the review, I would expect news about nanoencapsulation and bioactives. But most of the review is about biopolymer (cabohydrates) coating on meat. The authors should more focus on bioactives. There are a lot reviews about biodegradable package on food or meat, what's new the review could provide to readers? The authors should think about it.

Ans: Thank you very much for your suggestion, the role of encapsulation in the use of bioactive compounds in edible films or coatings has been added.

  1. I suggest the authors should add a part about the type of the bioactives encapsulated within edible coating. Which kind of bioactive are commonly used and how do they encapsulated by various methods.

Ans: Thank you very much for your comment, different bioactive compounds incorporated in edible films and coatings have been discussed in the paper.

  1. I suggest the authors add a schematic illustration to show the method of bioactive nano-encapsulated edible film making. I think there are different methods for the film making in literature. As a reader, I think this part will help the other researchers. 

Ans: It has been added in Figure.

Round 2

Reviewer 1 Report

The authors have incorporated all the suggestions in the article. 

Author Response

Thank you very much.

Reviewer 2 Report

The revised version is acceptable.

Author Response

Thank you very much.